# Strain-Insensitive Elastic Surface Electromyographic (sEMG) Electrode for Efficient Recognition of Exercise Intensities

**DOI:** 10.3390/mi11030239

**Published:** 2020-02-25

**Authors:** Daxiu Tang, Zhe Yu, Yong He, Waqas Asghar, Ya-Nan Zheng, Fali Li, Changcheng Shi, Roozbeh Zarei, Yiwei Liu, Jie Shang, Xiang Liu, Run-Wei Li

**Affiliations:** 1Faculty of Materials Science and Engineering, Kunming University of Science and Technology, Kunming 650093, China; tangdaxiu@nimte.ac.cn; 2CAS Key Laboratory of Magnetic Materials and Devices, Ningbo Institute of Materials Technology and Engineering, Chinese Academy of Sciences, Ningbo 315201, China; yuzhe@nimte.ac.cn (Z.Y.); yung_he@163.com (Y.H.); waqas@nimte.ac.cn (W.A.); zhengyanan@nimte.ac.cn (Y.-N.Z.); lifali@nimte.ac.cn (F.L.); changchengshi@nimte.ac.cn (C.S.); roozbeh.zarei@gmail.com (R.Z.); liuyw@nimte.ac.cn (Y.L.); 3Zhejiang Province Key Laboratory of Magnetic Materials and Application Technology, Ningbo Institute of Materials Technology and Engineering, Chinese Academy of Sciences, Ningbo 315201, China; 4Center of Materials Science and Optoelectronics Engineering, University of Chinese Academy of Sciences, Beijing 100049, China; 5Department of Mechanical Engineering, University of Engineering and Technology Taxila, Taxila 47050, Pakistan; 6Swinburne Data Science Research Institute, Swinburne University of Technology, Melbourne, VIC 3122, Australia

**Keywords:** elastic sEMG electrode, strain-insensitivity, electrode–skin impedance, skin-conformability, signal-to-noise ratio

## Abstract

Surface electromyography (sEMG) sensors are widely used in the fields of ergonomics, sports science, and medical research. However, current sEMG sensors cannot recognize the various exercise intensities efficiently because of the strain interference, low conductivity, and poor skin-conformability of their electrodes. Here, we present a highly conductive, strain-insensitive, and low electrode–skin impedance elastic sEMG electrode, which consists of a three-layered structure (polydimethylsiloxane/galinstan + polydimethylsiloxane/silver-coated nickel + polydimethylsiloxane). The bottom layer of the electrode consists of vertically conductive magnetic particle paths, which are insensitive to stretching strain, collect sEMG charge from human skin, and finally transfer it to processing circuits via an intermediate layer. Our skin-friendly electrode exhibits high conductivity (0.237 and 1.635 mΩ·cm resistivities in transverse and longitudinal directions, respectively), low electrode–skin impedance (47.23 kΩ at 150 Hz), excellent strain-insensitivity (10% change of electrode–skin impedance within the 0–25% strain range), high fatigue resistance (>1500 cycles), and good conformability with skin. During various exercise intensities, the signal-to-noise ratio (SNR) of our electrode increased by 22.53 dB, which is 206% and 330% more than that of traditional Ag/AgCl and copper electrode, respectively. The ability of our electrode to efficiently recognize various exercise intensities confirms its great application potential for the field of sports health.

## 1. Introduction

Nowadays, flexible biosensors have revolutionized the field of human health monitoring, by smart tracking of various signals, such as electromyograms (EMGs) [1,2,3,4,5,6,7,8,9,10,11,12,13,14], electrocardiograms (ECGs) [2,4,6,7,9,11,13,15,16,17,18], electroencephalograms (EEGs) [2,6,15] and pulse [3,4,9,13,19]. Among them, the flexible surface electromyography (sEMG) sensors have gained more attraction due to their vast applications in exoskeleton, intelligent prosthesis, and motion monitoring. The sEMG signal is generated by the excitation of the somatic nervous system, which alters the permeability of muscle fiber cells and then changes the resting potential of muscle cells to action potential. The potential difference between resting and action potential is called the sEMG signal. The strength of this sEMG signal depends on the strength of the muscle contraction. As the strength of muscle contractions increases, the number of participating muscle fibers and superimposed action potentials also increases, which eventually leads to an increase in the sEMG signal’s strength.

Generally, two types of sEMG electrodes are available on commercial scale, i.e., metal electrodes and Ag/AgCl electrodes. Metal electrodes (especially copper [12] or titanium [4,8]) exhibit excellent conductivity and provide continuous monitoring for a long time. However, these electrodes bear certain disadvantages such as poor adhesion to the human skin due to their high mechanical rigidity, high electrode–skin impedance, occurrence of noise due to relative displacement between skin and electrode, being uncomfortable to wear, and causing skin irritation and skin abrasions. Ag/AgCl electrodes are gel-type electrodes, which are famous for their good skin-conformability and lower electrode–skin impedance values. However, certain disadvantages are also associated with them, namely, poor strain-insensitivity, vulnerability to noise, evaporation of gel moisture, which makes the electrode ineffective and a disposable product [10,20]. 

In recent years, a new type of electrode called elastic sEMG electrodes have been rapidly developed for monitoring of sEMG signals. These electrodes are composite in nature and are prepared by mixing nano- or micro-scale conductive fillers in an elastic matrix. Conductive fillers mainly consist of carbon-based conductive materials (such as graphene [2,4,8,19,21,22,23,24,25] and carbon nanotubes [10,11,13,21,23,24,25,26]) and metal-based conductive materials (such as gold [1,5,9,11,16,17,18,25,27,28], silver [4,7,10,15,25,29,30,31], nickel [25,27,32,33], and copper [25,34,35,36]); while polydimethylsiloxane (PDMS) [7,9,11,16,17,18,19,21,22,24,25,27,28,29,30,31,36,37] and polyurethane (PU) [11,34,35] are two commonly used matrix materials of elastic sEMG electrodes. However, there exists a modulus mismatch between the solid conductive fillers and the elastic matrix (a difference of 5–7 orders of magnitude [38]), which limits the doping amount of conductive filler. In addition, strain interference during the body movement interrupts the conductive path made by fillers which seriously affects the conductive stability of the electrode. In general, the resistance of previously reported elastic electrodes lies in the range of ohms to megaohms [10,17,24,25], and it increases 0.3–10 times [18,21,27,28,29,30,34,35,36,39] when electrode is subjected to 30% tensile strain (the mean tension of the skin is about 25% [40]). This lowers the signal-to-noise ratio (SNR) of the sEMG signal and the signal varies minutely on the increase of exercise intensity. 

Therefore, in all the above-mentioned sEMG electrodes, the minute change occurring in signal strength is disturbed by noise, which is caused due to strain interference and relative motion between the skin and the electrode. Hence, the efficient recognition of various exercise intensities becomes difficult, especially for the athletes to determine whether their muscle exercise is up to standard and for doctors to efficiently monitor the muscle recovery of patients. Therefore, to recognize various exercise intensities, the development of an electrode with high conductivity, excellent strain-insensitivity, good skin-conformability, low electrode–skin impedance, and high SNR remains a challenge.

In this paper, we present a highly conductive, strain-insensitive, and low electrode–skin impedance elastic sEMG electrode, which consist of a three-layered structure (polydimethylsiloxane/galinstan + polydimethylsiloxane/silver-coated nickel + polydimethylsiloxane), i.e., (PDMS/LM + PDMS/MPs + PDMS). In the paper, polydimethylsiloxane, galinstan and silver-coated nickel are replaced by PDMS, LM and MPs respectively. The strain-insensitive bottom layer of the electrode performs a dual function of preventing LM leakage and of collecting sEMG charge from the human skin followed by transferring it to processing circuits via an intermediate layer. When characterized, our elastic sEMG electrode has shown strain-insensitive behavior with high conductivity, low electrode–skin impedance, good skin-conformability, and skin-friendliness at the same time. Besides this, our electrode also exhibits high SNR and is fully capable of recognizing the various exercise intensities, which confirms its great application potential for the field of sports health.

## 2. Materials and Methods 

### 2.1. Chemicals and Materials

PDMS (Sylgard 184, Dow Corning, Midland, MI, USA) and silver-coated nickel (MPs) (average diameter: ~14 μm, resistivity: ~0.55 mΩ·cm, Potters Industries Inc., USA) were purchased from the market. High-purity metals gallium, indium, and tin (99.99%, Beijing Founde Star Science and Technology Co., Ltd., Beijing, China) were mixed in the ratio of 68.5:21.5:10 by mass. Then, the above mixture was heated and stirred at 70 °C for 120 min to obtain LM galinstan (resistivity: ~0.294 mΩ·cm). 

### 2.2. Designing Strategy of Elastic sEMG Electrode

In order to efficiently recognize various exercise intensities, we prepared an elastic electrode that consists of a three-layered structure, i.e., a top layer, an intermediate layer, and a bottom layer (PDMS/LM + PDMS/MPs + PDMS). The design principle of the three-layered elastic sEMG electrode is shown in Figure 1. The electrode’s intermediate layer was made up of composite material (LM and PDMS), because of its unique properties: high electrical conductivity (1.34 × 10^3^ S·cm^−1^), large elongation at break (116.86%), and minute resistance fluctuation (4.305%) when stretched to 100% [41]. However, under strain conditions, elastomer-encapsulated LM leaked from the electrode surface and contaminated the skin. To cope with this problem, the intermediate composite layer was protected by sandwiching it between PDMS (top) and MPs + PDMS (bottom) layers. The top PDMS layer performed a dual function by simultaneously preventing the leakage of LM and sEMG charges. The bottom layer of MPs + PDMS also exhibited multifunctional characteristics of good electrical conductivity, good skin-friendliness, and strain-insensitivity. When a curable mixture of MPs and PDMS is subjected to a vertical magnetic field (B_vert._), it results in the formation of conductive paths in MPs, along the magnetic field’s direction. A small amount of such doping does not alter the elastic properties of the elastomer but gives considerable electrical conductivity in the vertical direction. For example, Majidi et al. [32] found that the electrical resistivity of magnetically cured MPs + PDMS composite (40% wt.) was only 0.03 Ω.m, and the elongation at break was close to 110%. Our bottom layer consists of vertically conductive MPs paths (always conductive in a direction opposite to applied strain), hence this property makes it strain-insensitive. The bottom layer collects the sEMG charge generated on the skin and transfers it to the back-end processing circuit via the intermediate layer. Beside this, the bottom layer also prevents LM leakage and ensures skin-friendliness. Therefore, by using such a three-layered structure, high conductivity, strain-insensitivity, and skin-friendly can be achieved simultaneously. If the thickness of the electrode layers is kept small, the electrode will also become skin conformable at the same time.

### 2.3. Preparation and Microstructure Characterization of Elastic sEMG Electrode

Figure 2a shows the schematic image of the three-layered electrode manufacturing steps. The process starts by pouring a mixture of MPs and PDMS in a polystyrene mold. After the removal of air bubbles, the mold was placed under vertical magnetic field (B_vert_.) and then, the bottom layer was cured at 70 °C for 45 min. The intermediate layer was prepared as described in previously published literature [42], which roller prints a stirred mixture of LM and PDMS (16:1 mass ratio), followed by curing at 70 °C for 1 h. After curing the intermediate layer, liquid PDMS was poured on its surface and then, the PDMS layer was cured at 70 °C for 45 min. The curing of the top PDMS layer completed the formation of the elastic sEMG electrode. Finally, the cured sample was peeled from the mold and then cut into the required rectangular shape. In these steps, the mass ratios of MPs and PDMS were an important parameter to decide the electrical property of the bottom layer. Therefore, we studied the effect of this mass ratio parameter on the resistivity of the bottom layer. The results are shown in Figure 2b. The longitudinal resistivity decreased significantly with the increase of mass ratios (i.e., MPs concentration). When the mass ratio reached 1:1, the longitudinal resistivity tended to be stable and, thereby, the optimum technology parameter was confirmed as 1:1 mass ratio to prepare the bottom layer. According to the four probes testing result of resistance profile (shown in Figure 2c), the surface resistivity of the three-layered electrode was lower than 0.2 mΩ/sq, which indicates its high electrical conductivity.

The cross-sectional microstructure of the prepared electrode was characterized by an optical microscope and scanning electron microscope (SEM), as shown in Figure 2d. The images clearly reveal the three functional layers of the electrode, which were designated by I, II, and III, respectively. From the optical microscope image, it can be found that the vertically arranged MPs are visible as a columnar structure in PDMS matrix of the bottom layer. The intermediate layer depicts that LM forms a 3D-Calabash Bunch of conductive networks in PDMS, as shown in the SEM image [41]. Meanwhile, the total thickness of the electrode was 200 µm, which enabled our electrode to have a good skin-conformance for collecting the weak electromyography signal (shown in Figure 2e).

### 2.4. Testing of Electromechanical, Electrode–Skin Impedance and sEMG Signal Collection

The stretching strain was applied by a universal material testing machine (Instron 5943, Instron Corp., Norwood, MA, USA) with a speed of 1.0–3.0 cm/min. The machine grips of Instron 5943 are a ceramic material, hence it was electrically isolated. The electrical property of the sample was measured by the DC(DC means direct current) current source (Keithley 6221, Keithley Instruments, Cleveland, OH, USA) and the nanovoltmeter (Agilent 34420A, Agilent Technologies, Loveland, CO, USA). During electromechanical testing, the electrode sample was connected to the instruments by silver wires. A photograph of the experiment is shown in Appendix A. The LM was smeared on the conductive interface between the sample and the silver wire, and then the PDMS was utilized to package the interface in order to realize a stable and highly conductive interface.

To measure the electrode–skin impedance, we parallelly mounted two of our electrodes on human skin. The distance between these two electrodes was kept at 1 cm, and impedance was tested by using an impedance analyzer (IM 3570, HIOKI, Ueda, Japan). The impedance tests were in the range of 20–400 Hz, since sEMG signals concentrate well at the range of 100–150 Hz [7,14].

The Delsys sEMG system was chosen to collect signals. In order to reduce the noise interference caused by leading wires, we replaced the insulated material of the top layer (PDMS) with the conductive material of the bottom layer (MPs + PDMS). The new electrode was cut into four elastic sEMG electrodes with a size of 1 cm × 0.5 cm to attach to the back-end circuit of the Delsys system (black module) by double-sided tape, as shown in Appendix A. These four electrodes were used as the reference electrodes and the test electrodes. As shown in Appendix A, the electrode and black module were fixed to the skin of the biceps brachii via acrylic tape. The collected signals were transmitted to the computer via the circuit’s Bluetooth and displayed in real time.

## 3. Results and Discussion

### 3.1. Temperature-Insensitivity and Strain-Insensitivity

Resistivity variation of the electrode at different temperatures is presented in Appendix A, in which the electrode showed good temperature stability within the temperature range of 5–75 °C (the temperature range of healthy people is 35–36.8 °C). At 25 °C, the resistivities of the electrode were 0.237 and 1.635 mΩ·cm in the transverse and longitudinal directions, respectively. Overall, 1.15% and 8.81% total change in resistivity was observed in the transverse and longitudinal directions, which is negligible.

Electrode elongation at break was found to be 144.48% (Appendix A), which confirms the high elasticity of our electrode. As skin strain generally remains less than 30%, we electromechanically characterized our electrode at 10%, 20%, and 30%. The electromechanical characterization of the elastic electrode is presented in Figure 3. When the electrode was mechanically strained, it showed good tensile recovery during the loading–unloading process, indicating small hysteresis (Figure 3a,b). The gauge factors (GFs) of our electrode in transverse direction and longitudinal direction were −0.234 and 0.265, respectively. This confirms the good strain-insensitivity of our electrode. The GF is defined as the ratio of resistance change (∆R/R_0_) to the applied stretching strain. During repeated loading–unloading operations at various strains, the resistivity change of electrode was lower than 8% regardless of the transverse or longitudinal direction, as shown in Appendix A. To evaluate the fatigue resistance, a cyclic stretching test was performed on the electrode, at a strain of 30%. Figure 3c shows that our electrode maintained its GF without any significant deterioration even when stretched up to 1500 cycles. The average GFs in the transverse direction and longitudinal direction are −0.252 and 0.269, respectively, which proves that our electrode is highly resistant to mechanical fatigue. To verify any LM leakage, we stretched our electrode at 30% stretching strain (100 times), followed by rubbing its surface with fingers. No sign of LM leakage was found, which confirms the excellent skin-friendliness of our electrode (Appendix A). In short, the three-layered sEMG electrode is insensitive to the stretching strain.

### 3.2. Electrode–Skin Impedance

The results of the impedance of our electrode are shown in Figure 4. Figure 4a shows the impact of the MPs:PDMS mass ratio on the impedance of the electrode, in which impedance decreases significantly with the increase of MPs concentration. Electrodes having 1:1 and 2:1 mass ratio have shown low electrode–skin impedance but when compared in terms of elasticity, the elasticity of 1:1 electrode was found to be superior to 2:1 electrode (Appendix A). That is why further characterizations are only performed on electrodes having 1:1 mass ratio. For the intermediate layer, we have selected 10:1 mass ratio (LM:PDMS), which gives good conductivity and negligible relative resistance variation, as stated in a previous study [41].

The effect of the electrode area on the impedance is shown in Figure 4b. For this purpose, we prepared three electrodes with an area of 1, 2, and 4 cm^2^, respectively. The electrode with the large cross-sectional area (4 cm^2^) exhibited low impedance. Overall, there exists an inverse relation between impedance and area of the electrode, i.e., impedance decreases significantly with the increase of the electrode’s area. This reveals that, by adjusting the electrode area, we can easily match the electrode–skin impedance with the impedance of the back-end circuit and high SNR signals can be achieved efficiently. However, it is very difficult to achieve high SNR for metal electrodes, because the compatibility of the electrode with human skin decreases with the increase of the electrode’s area.

To prove that our electrode exhibited both low electrode–skin impedance and high SNR, the impedance performance of one of our electrodes (S = 2 cm^2^) is compared with copper and Ag/AgCl electrodes of the same size (Figure 4c). The copper electrode was prepared in the lab, and the Ag/AgCl electrode was bought from the market. At 150 Hz, the impedance value of our electrode was only 47.23 kΩ, which is 20.39% higher than the Ag/AgCl electrode’s impedance but at the same time 535.18% lower than that of the metal electrode. The electrode–skin impedance of the Ag/AgCl electrode was slightly lower than that of our electrode because of the presence of moisture in it. With the passage of time, the moisture of Ag/AgCl electrode volatilizes, which leads to an increase in its impedance and simultaneous reduction of its sEMG signal efficiency (Appendix A). In addition, moisture evaporation also makes Ag/AgCl electrode unsuitable to be used for a long time. So, our elastic electrode is the best alternative solution, which is free from moisture, exhibits low electrode–skin impedance, and can be efficiently used for a long time. Figure 4d shows the impedance performance of the electrode under different tensile strains. The electrode–skin impedance remains stable under a tensile strain range of 0–25%. At 150 Hz, the electrode–skin impedance of our electrode changed only by 10%, while the impedance of laboratory prepared traditional electrode (Ag + PDMS) changed by 4.6 times (Appendix A), which confirms the high strain insensitivity of our elastic sEMG electrode.

### 3.3. Application Demo

To distinguish various exercise intensities, we employed our sEMG electrode on the gastrocnemius muscle of the right leg. Various exercise intensities were created by jumping from different heights (10, 20, 30, 40, and 50 cm) and the resulting processed signal is shown in Figure 5a. The sEMG signal was processed by IIR(IIR means infinite impulse response)filtering, removing background noise (mean absolute value), and root mean square error of data. After jumping from the various heights, the signal intensity of our electrode increased significantly as compared to both copper and Ag/AgCl electrodes. Figure 5b shows the sEMG electrode attached on the gastrocnemius muscle of the right leg. In order to quantitatively compare the degree of enhancement of sEMG signals, we used the following equation:
(1)$$SNR=20\;log(\frac{V_{s}}{V_{n}})$$


In the above equation, V_s_ shows the maximum output voltage and V_n_ corresponds to noise voltage. The resultant SNR values of the electrodes are shown in Figure 5c. The SNR of our electrode’s signal increased by 22.53 dB, during jumping from the height of 10 to 50 cm. However, the SNR of the metal and Ag/AgCl electrodes increased only by 6.82 and 10.91 dB, respectively. Therefore, within the same height range (i.e., 10–50 cm), the SNR of our electrode increased more than 206% and 330%, when compared with traditional Ag/AgCl and copper electrodes, respectively. This fully demonstrates the advantage of our sEMG electrode for evaluating exercise intensity. Moreover, we placed our electrode on the flexor carpi muscle of the arm, to record the various gestures of the human hand. Our electrode successfully recognized all the gestures, as shown in Appendix A.

## 4. Conclusions

In this paper, we presented a highly conductive, strain-insensitive, skin-conformable, and skin-friendly elastic sEMG electrode, which consists of a three-layered structure (PDMS/LM + PDMS/MPs + PDMS). When characterized, our electrode showed high conductivity (0.237 and 1.635 mΩ·cm resistivity in the transverse and longitudinal directions, respectively), low electrode–skin impedance (47.23 kΩ at 150 Hz), excellent strain-insensitivity (10% change of electrode–skin impedance within a 0–25% strain range), high fatigue resistance (>1500 cycles), and good conformability with skin. The high conductivity of the electrode is attributed to the presence of LM and MPs inside. The strain-insensitivity is produced because of LM’s fluidity inside PDMS and the presence of MPs’ conductive paths in a direction opposite to applied stretching strain. The encapsulation of the intermediate layer prevents the LM leakage, ensures excellent skin-friendliness, and small overall thickness which makes our electrode skin-conformable at the same time. The low electrode–skin impedance, obtained due to high conductivity and excellent good conformability, reduces the noise and increases the SNR of the signal. When practically demonstrated, the signal’s SNR was retained due to strain-insensitivity, which makes our electrode a special candidate for human health monitoring in sports and medical fields.

## Figures and Tables

**Figure 1 micromachines-11-00239-f001:**
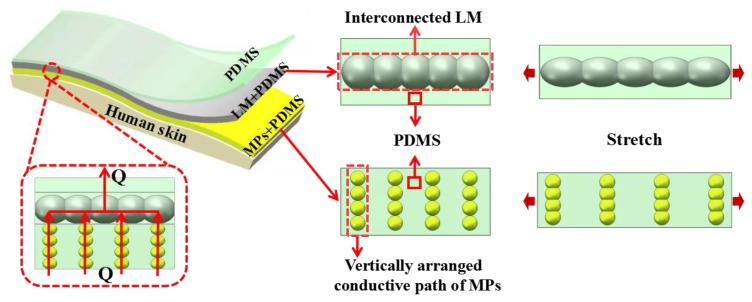
Design principle of three-layered elastic surface electromyography (sEMG) electrode (polydimethylsiloxane (PDMS)/galinstan (LM) + PDMS/silver-coated nickel (MPs) + PDMS). Sensor layers are in the configuration: top layer (PDMS)/intermediate layer (LM + PDMS)/bottom layer (MPs + PDMS). Inset, figure showing the flow of the sEMG charge generated from the skin.

**Figure 2 micromachines-11-00239-f002:**
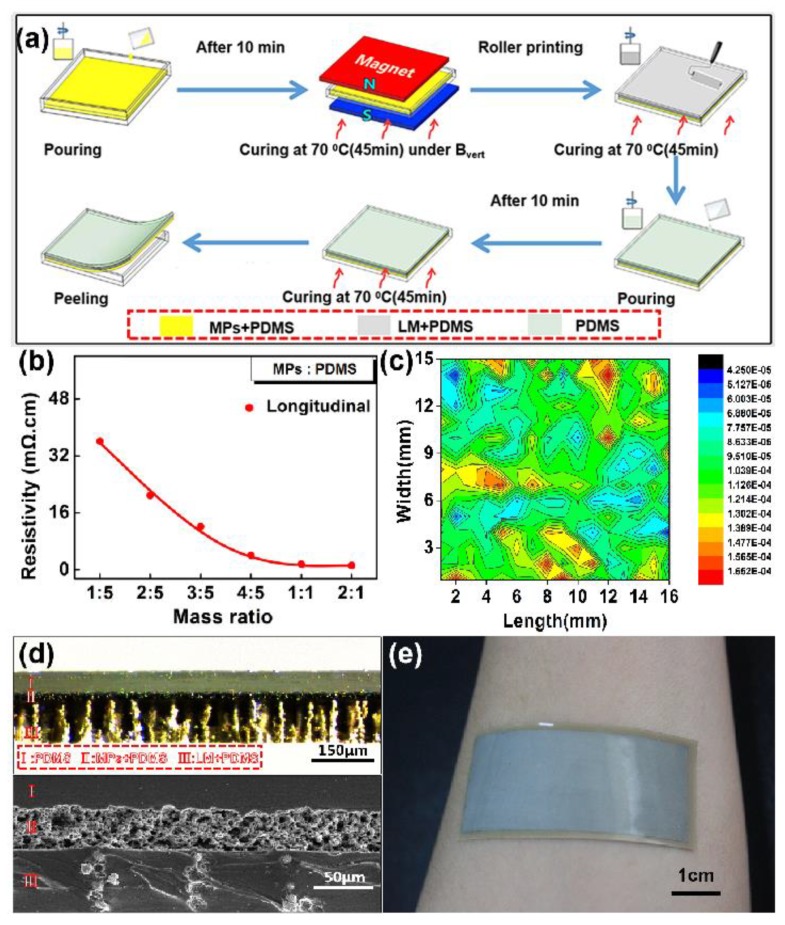
Fabrication of three-layered elastic sEMG electrode: (**a**) Schematic illustration for the fabrication of elastic sEMG electrode. (**b**) Change in longitudinal resistivity of bottom layer (MPs + PDMS) under different mass ratios. (**c**) The mapping images of surface resistivity of elastic electrode. (**d**) The cross-sectional images of electrode: optical microscope image (top) and scanning electron microscope (SEM) image (bottom). (**e**) A photograph of the electrode sample conformed to skin.

**Figure 3 micromachines-11-00239-f003:**
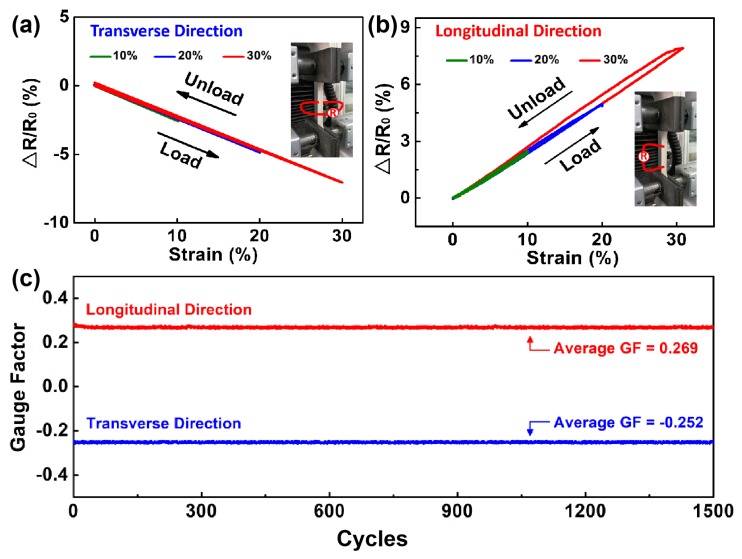
Electromechanical characterization of the three-layered sEMG electrode. Resistance in different directions as a function of the loading and unloading of various strains: (**a**) transverse resistivity and (**b**) longitudinal resistivity. **(c)** Gauge factors (GFs) as a function of the stretching cycle. The stretching strain was 30%.

**Figure 4 micromachines-11-00239-f004:**
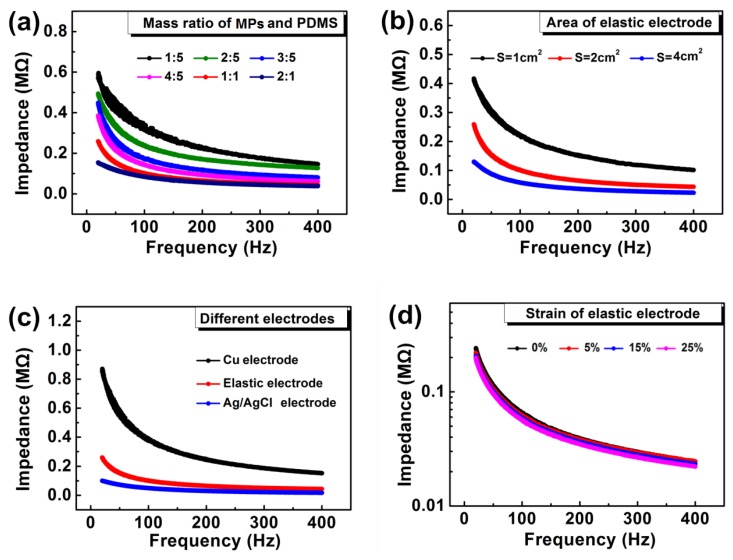
Electrode–skin impedance tests of sEMG electrode: (**a**) Impedance variation w.r.t change in MPs:PDMS mass ratio. (**b**) Effect of electrode area on impedance of electrode. (**c**) Impedance comparison of various electrodes. Cross-sectional area of all electrodes is the same (i.e., 2 cm^2^). (**d**) Impedance performance of our elastic electrode (2 cm^2^) under different tensile strains.

**Figure 5 micromachines-11-00239-f005:**
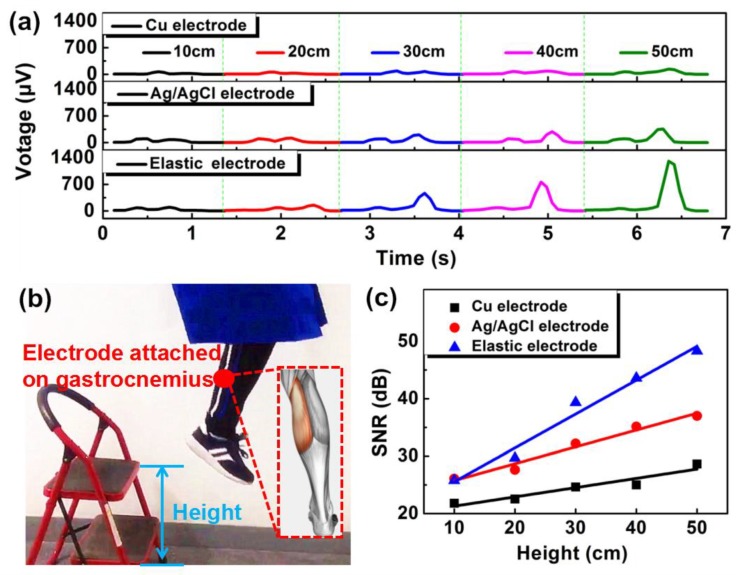
Application demo of our sEMG electrode. (**a**) Comparison of sEMG signals obtained after jumping from different heights (10, 20, 30, 40, and 50 cm). (**b**) sEMG electrode attached on gastrocnemius muscle of the right leg. Inset shows the gastrocnemius muscle of human leg. (**c**) SNR values comparison of various sEMG signals.

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
