# Peer review of "Strain-Insensitive Elastic Surface Electromyographic (sEMG) Electrode for Efficient Recognition of Exercise Intensities"

_micromachines, 2020, doi:10.3390/mi11030239_

Round 1

Reviewer 1 Report

The authors presented a fabrication method for strain-insensitive elastic electrode. The presented results are valuable. However, it is suggested that the supplementary materials should not be included in main texts (e.g. figure S1, figure S2a, figure S4 and their descriptions). If necessary, the authors should put these results into main texts and make some rearrangements.  

Reviewer 2 Report

Overall the work presents a complete study of sEMG electrode design, without any obvious methodology issues and is an important contribution to sEMG electrode research.

There are a few corrections to consider:

It's confusing to the have figures from the supplementary material references in the main text, as readers will naturally look for the figure in the text (for example Figure 1), and see Figure S1 instead. Since there's not a lot of supplementary material, just integrate it into the main paper as at the moment it just makes the work confusing to read through.

For example, Line 128-129: Figure S1 shows the resistivity of bottom layer measured in longitudinal and transverse directions.

The reader will naturally look above to Figure 1, not S1 at the end of the paper. Since Figure 1 doesn't show the resistivity (there is no measurement here), but rather shows the arrangement of the materials it would be easier just to include S1 in the main text.

As another example, Figure 3 and S4 are exactly the same, so why not just keep everything in Figure 3 and remove S4.

The electro-mechanical characterization set up is not described. There needs to be a picture showing the experimental setup (similar to Figure S3 and S5). In particular show how the material is connected to electrodes (via a flat electrode, conductive epoxy?), how it was mounted in the tensile machine (any issue with electrical isolation from the machine grips), and the details of the machine. Also the extension rate of the test in mm/sec is needed for the quasi-static loading and fatigue tests. This information should be put in the 2. Materials and Methods section.

Figure 3c could actually be replaced with calculating and plotting the gauge factor over the test cycles, so you would have one curve as opposed to the cycles which are more difficult to get information from directly. It would be interesting to know if there's any influence of strain rate on the electro-mechanical behavior, but that's better put in a future study.

Line 195 and Figure 4 description, "SEMG" should be changed to "sEMG"

In 3.3 Application Demo how were the sEMG electrodes attached to the gastrocnemius muscle? Also for the gesture demo, how were the electrodes attached and how was the signal recorded (what was the hardware setup)?
